# DHER: Hindsight Experience Replay for Dynamic Goals

**Meng Fang,**[*] **Cheng Zhou, Bei Shi, Boqing Gong, Jia Xu, Tong Zhang**
Tencent AI Lab

## Abstract

Dealing with sparse rewards is one of the most important challenges in reinforcement learning (RL), especially when a goal is dynamic (e.g., to grasp a moving object). Hindsight experience replay (HER) has been shown an effective solution to handling sparse rewards with fixed goals. However, it does not account for dynamic goals in its vanilla form and, as a result, even degrades the performance of existing off-policy RL algorithms when the goal is changing over time.

In this paper, we present Dynamic Hindsight Experience Replay (DHER), a novel approach for tasks with dynamic goals in the presence of sparse rewards. DHER automatically assembles successful experiences from two relevant failures and can be used to enhance an arbitrary off-policy RL algorithm when the tasks' goals are dynamic. We evaluate DHER on tasks of robotic manipulation and moving object tracking, and transfer the polices from simulation to physical robots. Extensive comparison and ablation studies demonstrate the superiority of our approach, showing that DHER is a crucial ingredient to enable RL to solve tasks with dynamic goals in manipulation and grid world domains.

## 1 Introduction

Deep reinforcement learning has been shown an effective framework for solving a rich repertoire of complex control problems. In simulated domains, agents have been trained to perform a diverse array of challenging tasks (Mnih et al., 2015; Lillicrap et al., 2015; Duan et al., 2016). In order to train such agents, it is often the case that one has to design a reward function that not only reflects the task at hand but also is carefully shaped (Ng et al., 1999) to guide the policy optimization. Unfortunately, many of the capabilities demonstrated by reward engineering are often limited to specific tasks. Moreover, it requires both RL expertise and domain-specific knowledge to reshape the reward functions. For situations where we do not know what admissible behavior may look like, for example, using LEGO bricks to build a desired architecture, it is difficult to apply reward engineering. Therefore, it is essential to develop algorithms which can learn from unshaped and usually sparse reward signals.

Learning with sparse rewards is challenging, especially when a goal is dynamic. Dynamic goals are common in games and planning problems, often addressed using reward shaping or search (Kaelbling, 1993; Mnih et al., 2015; Di Rocco et al., 2013). However, the difficulty posed by a sparse reward is exacerbated by the complicated environment dynamics in robotics (Andrychowicz et al., 2017). For instance, system dynamics around contacts are difficult to model and induce sensitivity in the system to small errors. Many robotic tasks also need executing multiple steps successfully over a long horizon, involve enormous search space, and require generalization to varying task instances. Policy gradient methods are breakthroughs in the challenging environments, such as PPO (Heess et al., 2017; Schulman et al., 2017), ACER (Wang et al., 2016), TRPO (Schulman et al., 2015) and so on. They are used in environments, where an agent tries to reach a target, learns to walk, runs, and so on. Recently, sampling-efficient learning is introduced and demonstrates a significant increase in performance for off-policy actor-critic DQN (Mnih et al., 2015) and DDPG (Lillicrap et al., 2015) algorithms. Hindsight experience replay (HER) is very effective for improving the performance of off-policy RL algorithms in solving goal-based tasks with sparse rewards (Andrychowicz et al., 2017). Similar to UVFA (Schaul et al., 2015a), it takes a goal state as part of input. However, it

---

[*]Correspondence to: Meng Fang <mfang@tencent.com>.

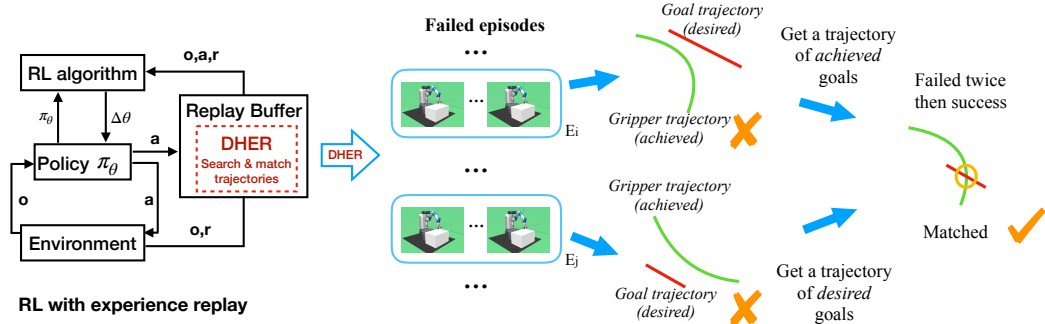

Figure 1: The framework of DHER. DHER is a kind of experience replay method. It searches relevant failed experiences and then assembles them into successful experiences.

assumes the goal is fixed. As a result, this assumption actually impedes the learning of RL agents in the environments of moving goals.

In this paper, we address this challenge with a new method, Dynamic Hindsight Experience Replay (DHER), for accomplishing tasks with moving goals. We follow the multi-goal setting in UVFA (Schaul et al., 2015a) and HER (Andrychowicz et al., 2017). It assumes that the goal being pursued does not influence the environment dynamics. We also need to have the knowledge of goal similarity. For example, in manipulation or grid world domains, we can use Euclidean distance between positions to measure the goal similarity. HER turns a failed episode to a success by composing a new task whose goal is achieved by that episode. Our idea allows an agent to learn from the failure one step further than HER: the agent not only sets a new goal but also hallucinates how to reach the original goal from the new one. Take playing frisbee for instance. When an agent jumps to catch the frisbee and yet misses it, the agent receives no positive feedback under the sparse reward setting. Using HER, the agent could set the end of its episode as the new goal — position of the frisbee; with DHER, however, the agent finds a trajectory from its past experiences as the imagined path of the frisbee, and thereby extrapolates towards the original goal.

In particular, we do the following for DHER. To finish the tasks with dynamic goals needs to explore experience and understand multiple goals. DHER uses replay buffers to allow the agent to learn from a couple of failures by assembling new 'experience' from different episodes. The proposed method retrieves memories to find the connection between the experience of different episodes. It largely improves the sample efficiency in dynamic goal task settings. More importantly, this strategy makes it possible to learn in the setting that both sparse rewards and dynamic goals exist.

We evaluate our method along with the state-of-the-art baselines on new environments and manipulation tasks, which have sparse rewards and dynamic goals. Our results demonstrate that DHER is clearly better than others for these tasks. We also transfer policies trained in our simulation based on DHER to a physical robot and show that DHER can be applied to solving real-world robotics problems.

We summarize our main contributions as follows: (1) We demonstrate that, both in simulation and real worlds, DHER succeeds in continuous control with a moving target. To our knowledge, this is the first empirical result on manipulation tasks that demonstrates model-free learning methods can tackle tasks of this complexity. (2) We show that with assembling new experience from two failures, the sample complexity can be reduced dramatically. We attribute this to global knowledge learning in a set of failed experience which breaks the constraint of local one-episode experience towards more robust strategies. (3) We design and implement a set of new environments and continuous tasks with dynamic goals, which would be of interest to researchers at the intersection of robotics and reinforcement learning.[1]

---

[1]Our code and environments are available at `https://github.com/mengf1/DHER`.

## 2 RELATED WORK

Recent works in deep RL have shown impressive results in different domains, such as games (Bellemare et al., 2013; Silver et al., 2016), simulated control tasks (Brockman et al., 2016) and so on. There have been several proposed RL methods for playing Atari games, including DQN (Mnih et al., 2015), UVFA (Schaul et al., 2015b) and so on. UVFA trains a single neural network approximating multiple value functions for state and goal. For the continuous control, DDPG (Lillicrap et al., 2015) is a popular actor-critic algorithm that has shown impressive results in continuous control tasks. Dynamic goals appear in games and planning, often addressed by rewards shaping or search (Kaelbling, 1993; Mnih et al., 2015; Di Rocco et al., 2013). When the rewards are sparse, there is few work studying the dynamic goals to the best of our knowledge.

Curriculum learning is also used for reinforcement learning scenarios. The idea is that solving easier problems first has advantages to learn more complex goals later and thus that learning can be optimized by presenting the problems in an optimal order, a curriculum (Bengio et al., 2009). Narvekar et al. (2017) and Florensa et al. (2018) proposed methods to automatically produce subtasks or subgoals for a given target. Narvekar et al. (2017) produce subtasks according to predefined tasks of a given domain problem. Florensa et al. (2018) use a Generative Adversarial Network (GAN) to produce goals with different difficulties. Different from these methods, our approach produces a series of goals for an episode from failed experience.

Experience replay is an important technique and introduced to break temporal correlations by mixing more and less recent experience for updating policies (Lin, 1992). It was demonstrated for its efficiency in DQN (Mnih et al., 2015). Prioritized experience replay improves the speed of training by considering prioritizing transitions in the replay buffer (Schaul et al., 2015b). HER considers modifying experience in the replay buffer for continuous control (Andrychowicz et al., 2017). By contrary, our approach assembles successful experience from two failures. Comparing with these methods, which do not consider dynamic goals, our approach uses a series of goals to assemble successful experience. As a result, our method is able to accomplish the tasks with sparse rewards and dynamic goals.

## 3 METHODOLOGY

We first review how HER works (Andrychowicz et al., 2017), followed by details of the proposed DHER for dealing with dynamic goals.

HER is a simple and effective method of manipulating the replay buffer used in off-policy RL algorithms that allows it to learn policies more efficiently from sparse rewards. It assumes the goal being pursued does not influence the environment dynamics. After experiencing an episode $\{s_0, s_1, \cdots, s_T\}$, every transition $s_t \rightarrow s_{t+1}$ along with the goal for this episode is usually stored in the replay buffer. Some of the saved episodes fail to reach the goal, providing no positive feedback to the agent. However, with HER, the failed experience is modified and also stored in the replay buffer in the following manner. The idea is to replace the original goal with a state visited by the failed episode. As the reward function remains unchanged, this change of goals hints the agent how to achieve the new goal in the environment. HER assumes that the mechanism of reaching the new goal helps the agent learn for the original goal.

### 3.1 DYNAMIC GOALS

Dynamic goals are not static and change at every timestep. We follow the multi-goal setting of Andrychowicz et al. (2017). The goals are part of the environment and do not influence the environment dynamics. We also assume that a dynamic goal $g_t \in \mathcal{G}$ moves by following some law $g_t = g(t; \gamma)$, where $\gamma$ parameterizes the law (e.g., acceleration in Newton's law of motion), and yet its underlying moving law is unknown to the agent.

Moreover, we need to have some basic knowledge of goals, i.e., the measure of goal similarity on $\mathcal{G}$. We assume that $g_t \in \mathcal{G}$ corresponds to some predicate $f_{g_t} : S \rightarrow \{0, 1\}$ and that the agent's goal is to achieve any state $s$ that satisfies $f_{g_t}(s) = 1$. We use $S = \mathcal{G}$ and define $f_{g_t}(s) := [s = g_t]$, which can be considered as a measure of goal similarity between $g_t$ and $s$. The goals can also specify only some properties of the state. Take manipulation tasks for instance: $\mathcal{G} = \mathbb{R}^3$ corresponds to the 3D

---

**Algorithm 1** Dynamic Hindsight Experience Replay with Experience Assembling

---

**Require:** an off-policy RL algorithm $\mathbb{A}$, replay buffer $R$, a reward function $r$

1: Initialize $\mathbb{A}$ and replay buffer $R$
2: **for** episode $= 1, 2, \cdots, M$ **do**
3:      Sample an initial goal $g_0$ and an initial state $s_0$
4:      **for** $t = 0, \cdots, T - 1$ **do**
5:          Sample an action $a_t$ using the behavioral policy from $\mathbb{A}$:
6:          $a_t \leftarrow \pi(s_t|g_t)$
7:          Execute the action $a_t$ and observe a new state $s_{t+1}$ and a new goal $g_{t+1}$
8:      **end for**
9:      **for** $t = 0, \cdots, T - 1$ **do**
10:         $r_t := r(s_t, a_t, g_{t+1})$
11:         Store the transitions $(s_t|g_t, a_t, r_t, s_{t+1}|g_{t+1})$ in $R$    (Standard experience replay)
12:      **end for**
13:      Collect failed episodes to $\mathcal{E}$
14:      **for** $E_i \in \mathcal{E}$ **do**
15:         Search another $E_j(i \neq j) \in \mathcal{E}$ where $g_{i,p}^{ac} = g_{j,q}^{de}$
16:         **if** $E_j \neq \varnothing$ **then**
17:            Clone a goal trajectory $\{g_0', \cdots, g_m'\}_{m=\min\{p,q\}}$ in which $g_t' = g_{j,q-m+t}^{de}$ from $E_j$
18:            **for** $t = \{0, \cdots, m - 1\}$ **do**
19:               $r_t' := r(s_{i,p-m+t}, a_{i,p-m+t}, g_{t+1}')$
20:               Store the transition $(s_{i,p-m+t}|g_t', a_{i,p-m+t}, r_t', s_{i,p-m+t+1}|g_{t+1}')$ in $R$    (DHER)
21:            **end for**
22:         **end if**
23:      **end for**
24:      **for** $t = 1, \cdots, N$ **do**
25:         Sample a minibatch $B$ from the replay buffer $R$
26:         Optimize $\mathbb{A}$ using the minibatch $B$
27:      **end for**
28: **end for**

---

positions of an object $s^{\text{obj}}$ about which the observation could include additional properties of the object. For a more concrete example, consider pushing a block towards a moving target position. The success of a task is defined as $f(s_t, g_t) = \mathbf{1}_{\text{condition}}(\|s_t^{\text{obj}} - g_t\| \leq \epsilon)$, where $s_t^{\text{obj}}$ is the position of the object in the state $s_t$ and $\epsilon$ denotes a tolerance by the environment. $\mathbf{1}_{\text{condition}}$ is an indicator function. The agent aims to achieve any state $s_t$ that satisfies $f(s_t, g_t) = 1$. It receives a sparse reward $r_t := r(s_t, a_t, g_{t+1}) = -\mathbf{1}_{\text{condition}}(f(s_{t+1}, g_{t+1}) \neq 1)$ upon making an action $a_t$.

It is worth discussing the main difference between the implications of the dynamic goals and the static ones. Expressing a static goal in the following way, $g_t^{\text{static}} = g^{\text{static}}, \forall t$, highlights the key challenge of dealing with the dynamic goal. Namely, the agent has no access to the underlying law of the dynamic goal in our setting, whereas the law of being static is known to the agent in Andrychowicz et al. (2017). In other words, the agent has no clue at all how to construct a new dynamic goal that is admissible by the environment.

### 3.2 Dynamic Hindsight Experience Replay

At the first glance, we shall compose a new dynamic goal $g^{\text{dynamic}} = \{s_{t_0}^{\text{obj}}, s_{t_1}^{\text{obj}}, \cdots, s_{t_{T'}}^{\text{obj}}\}$ from a failed episode $s_0, s_1, \cdots, s_T$ in order to apply HER (Andrychowicz et al., 2017) to our problem setting. However, per the discussion above, this new dynamic goal $g^{\text{dynamic}}$ may be inadmissible by the environment, leading to no positive feedback to the agent at all.

We tackle the challenge by drawing the following two observations. One is that many episodes fail in the replay buffer, implying that the agent can actually build a new goal upon more than one episodes. The other is that the more failed experience the agent has, the more possible for the agent to use the connection between achieved goals in an episode and desired goals in some other episode. Take the example of a moving object that the agent must reach, desired goals are the positions of

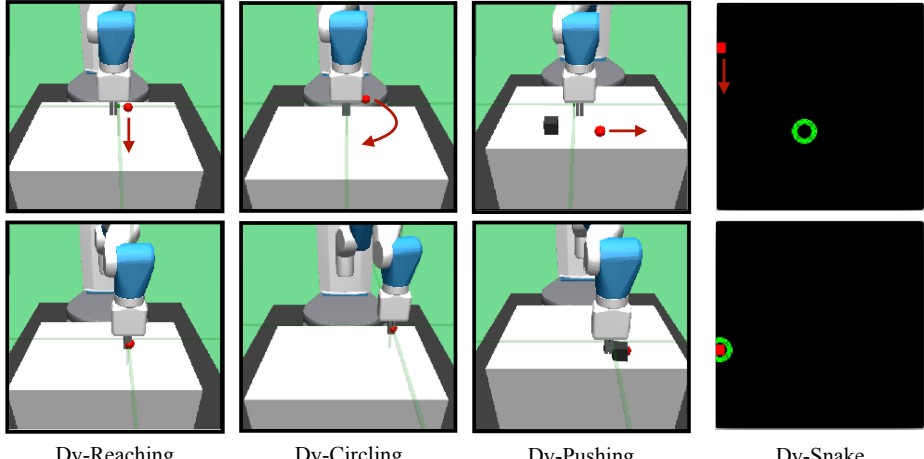



Dy-Reaching       Dy-Circling       Dy-Pushing       Dy-Snake



Figure 2: The proposed tasks with dynamic goals (red objects). Arrow indicates the movement of a goal. The first row indicates initial states. The second row indicates final states.

the moving object and achieved goals are the positions of a gripper (controlled by the agent). There may exist some positions that both the object and the gripper have ever reached respectively.

After experiencing some episode $s_0, s_1, \cdots, s_T$, we store in the replay buffer every transition $s_t \rightarrow s_{t+1}$ for this episode, defined as $(s_t, a_t, r_t, s_{t+1})$, where $s_t$ indicates a state $s_t$ at timestep $t$, and $a_t$ indicates an action and $r_t$ indicates a reward. Thus before $t$, there are a series of records $\{(s_0, a_0, r_0, s_1) \cdots, (s_t, a_t, r_t, s_{t+1})\}$. A state consists of three parts: observation $o_t$, desired goal $g_t^{de}$ and achieved goal $g_t^{ac}$, define as $s_t = \langle o_t, g_t^{ac}, g_t^{de} \rangle$, where normally $g_t^{de} = g_t$ and $g_t^{ac}$ indicates goals that the agent has achieved.

We reuse the failed experience from the replay buffer with inverse simulation to create successful rewards for the agent, as shown in Algorithm 1 (lines 13-23). Our inverse simulation contains two main steps: First, given a failed episode, for its achieved goal trajectory, we try to find a desired goal trajectory from other episodes that could match it (line 15). Second, we assemble a new episode by matching the achieved goal trajectory of the given episode to the desired goal trajectory of the founded one (lines 17-21). Let $g_{i,q}^{ac}$ indicate the achieved goal of the agent at timestep $q$ in episode $i$ and $g_{j,p}^{de}$ indicate the desired goal at timestep $p$ in episode $j$. Given a set of failed experience $\{E_0, E_1, E_2, \cdots\}$, we search and draw two failed episodes $E_i$ and $E_j$ ($i \neq j$), where $\exists i, j, p, q$, s.t. $g_{i,p}^{ac} = g_{j,q}^{de}$. If we find two such failed episodes, we combine the two experience by replacing the desired goals in $E_i$ by $\{g_{j,t}^{de}\}$, where $j$ indicates $E_j$ and $t \leq \min\{p, q\}$. Based on this, we end up assembling a new experience $E_i'$ based on $E_i$ with a new "imagined" goal trajectory $\{g_{j,0}^{de}, \cdots, g_{j,t}^{de}\}$ where $t \leq \min\{p, q\}$.

More details of the complete RL+DHER method are shown in Algorithm 1. Unlike HER, our DHER needs to search all failed experiences to compose a "imagined" goal trajectory. Hence, the efficiency of searching the memory is important. In our implementation, we use two hash tables to store the trajectories of achieved goals and desired goals, respectively.

## 4 EXPERIMENT

We run extensive experiments to examine the proposed DHER for moving goals and compare it with some competing baselines. We first introduce the environments and tasks that we want to address, followed by the experimental results. Demo videos from our experiments are available at https://sites.google.com/view/dher.

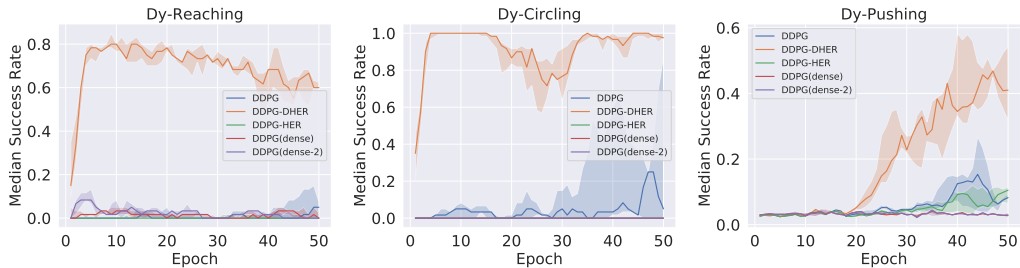

(a) Dy-Reaching with a straight-line goal.  (b) Dy-Circling with a goal that moves in a circle.  (c) Dy-Pushing with a straight-line goal.

Figure 3: Results on different environments.

## 4.1 ENVIRONMENTS

Whereas grasping moving targets are fairly common in robotic applications, there are rarely existing environments featuring dynamic goals. We modify the robotic manipulation environments created by OpenAI (Brockman et al., 2016) for our experiments. As shown in Figure 2, we assign certain rules to the goals so that they accordingly move in the environments while an agent is required to control the robotic arm's grippers to reach the goal that moves along a straight line (Dy-Reaching), to reach the goal that moves in a circle (Dy-Circling), or to push a block to the goal that moves along a straight line (Dy-Pushing). In addition, we also develop a new GREEDY SNAKE environment (Dy-Snake), in which the goal moves from one discrete cell to another. The greedy snake aims to reach the goal (and eat it). No matter what actions are taken by the agent, the underlying rule of changing the goals' positions remain the same. We assume that the law of the goal's motion is unknown to the agent. More details of the tasks are described in the appendix.

For the first three tasks, we follow the basic settings of OpenAI robotics environments (Brockman et al., 2016). States are read from the MuJoCo physics engine. An observation consists of relative positions of the object and the target (grippers are blocked) Goals are positions in the 3D world coordinate system with a fixed tolerance (we use $\epsilon = 0.01$ for the tolerance). Note the goals are able to move in our tasks. The velocity we use is $v = 0.011$. 1 epoch indicates 100 episodes. The start positions of the goals are randomly chosen. Rewards are binary and sparse: $r(s_t, a_t, g_t) = -\mathbf{1}_{\text{condition}}(|s_{t+1}^{\text{obj}} - g_{t+1}| \geq \epsilon)$ where $s_{t+1}$ and $g_{t+1}$ are respectively the state and dynamic goal after the execution of the action $a_t$ in the state $s_t$. Two positions are overlapped if they are close within the tolerance $\epsilon$. We use 3-dimensional actions, in which three dimensions correspond to the desired relative gripper position at the next timestep.

For the last task, it is very similar to the standard snake game. There is a 2D plane grid whose size is $30 \times 40$. A snake and a goal (food) both move in this grid. We use a $1 \times 1$ square as the body of the snake and do not allow the snake to grow any longer as it moves. The states of the system are represented by using the positions of the snake and food. Goals are the positions of the food. The snake has to move itself such that it resides in the same cell as the food at a certain timestep. The velocity of the goal is set to $(0, 1)$. The start positions of the goal are set randomly in different episodes of the game, so are the start positions of the snake. Rewards are binary and sparse: $r(s_t, a_t, g_{t+1}) = -\mathbf{1}_{\text{condition}}(|s_{t+1}^{\text{snake}} - g_{t+1}| \neq 0)$ where $s_{t+1}$ and $g_{t+1}$ are the environment's state and goal after the agent executes action $a_t$ in the state $s_t$. Observations are represented by the positions of the snake and food. We also add to the state the distance between the snake and food. Actions allowed in the game are the following: move up, move down, move left, and move right, for one cell per timestep.

## 4.2 BASELINES

For the first three tasks which call for continuous control, we consider two competing baselines:[2]

---

[2]https://github.com/openai/baselines

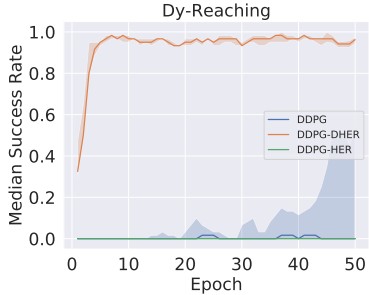

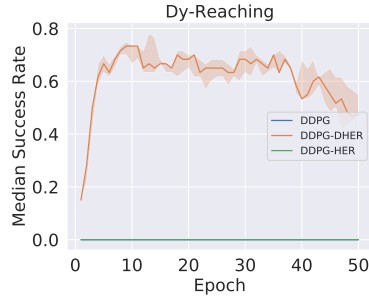

Figure 4: Low velocity: 0.001.

Figure 5: High velocity: 0.016.

- DDPG, which is a model-free RL algorithm for continuous control (Lillicrap et al., 2015). It learns a deterministic policy by using a stochastic counterpart to explore in the training.
- DDPG + HER, which improves the replay buffer of DDPG by the hindsight experience replay (Andrychowicz et al., 2017).
- DDPG (dense), which employs the negative distance $(-d)$ as dense rewards.
- DDPG (dense-2), which employs the negative distance $(-d)$ as dense rewards if $d \geq \epsilon$. However, if $d < \epsilon$ (i.e., success) it uses $(-d + 1.0)$ as rewards. $1.0$ is a bonus.

For the last task of discrete control, we use the following baselines in the experiments: DQN (Mnih et al., 2015) and DQN + HER, which uses HER to enhance the replay in DQN.

- DQN, which is a powerful model-free RL algorithm for discrete action spaces (Mnih et al., 2015).
- DQN + HER, which uses HER to enhance the replay in DQN.
- DQN (dense), which uses the negative distance $(-d)$ as dense rewards.
- DQN (dense-2), which uses the negative distance $(-d)$ as dense rewards. However, if $d = 0.0$ (i.e., success) it uses $(-d + 1.0)$ as rewards instead. $1.0$ is a bonus.

## 4.3 COMPARISON RESULTS ON THE ROBOTIC ENVIRONMENTS

For the continuous control, we present three sets of comparison results in Figure 3 for the first three tasks, respectively. Consistently, the results show our DHER algorithm outperforms the others. The two baselines are not able to catch up even after we train them for thousands of iterations. Vanilla DDPG is slightly better than the version with HER. HER does not benefit DDPG in these tasks because the goals in HER are fixed, fundamentally misleading the agent in the attempt of solving the tasks with dynamic goals. The results of DDPG (dense) and DDPG (dense-2) suggest that even the dense rewards do not work well as they are agnostic to the task of interest. A good reward shaping may give rise to better performance by carefully tuning it for the task of dynamic goals.

Comparing Figure 3a and Figure 3b with Figure 3c, we find that DHER learns faster in Dy-Reaching and Dy-Circling than Dy-Pushing probably because Dy-Reaching and Dy-Circling are easier tasks than Dy-Pushing. In Dy-Pushing, all the algorithms take a fairly big amount of time to explore without receiving any positive feedback. However, the more failed experiences the agent encounters, the better change our algorithm is able to identify relevant episodes from them for assembling useful dynamic goals. As a result, DHER is able to pick up the momentum and learns faster and better than the baselines after a certain point. In Figure 3a, the performance decreases a little. The reason may be that as successful experience increases, some assembled experience is inconsistent with these successful experience.

### 4.3.1 COMPARISON USING DIFFERENT VELOCITIES OF GOALS

To show the performance of our method on more complex tasks with different velocities, we study different methods in Dy-Reaching environment as shown in Figures 4 and 5 with the same physical

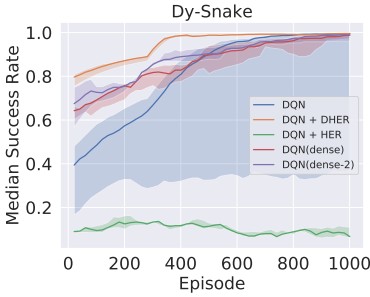

Figure 6: Snake with dynamic goals.

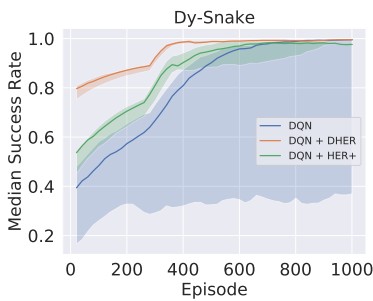

Figure 7: Special: HER+ knows how goals move.

properties as the previous experiments. Overall the results show our method is much better than DDPG and HER. As a reminder, the threshold for calculating rewards we used is $0.01$. In Figure 4, the task becomes an easier task because of the slower velocity. It shows that our method quickly achieves to a good result around $5$ epoch. Comparing with Figure 3a, it shows that the performance with $v = 0.001$ is better than the performance with $v = 0.011$ and get $15\%$ improvements. However, it also shows the performance with $v = 0.016$ is worse than the performance with $v = 0.011$. Both DDPG and DDPG+HER failed when $v = 0.016$ and their performance is $0$. This performance is consistent because the task becomes more difficult when the velocity increases.

### 4.4 COMPARISON ON THE DYNAMIC SNAKE ENVIRONMENT

In the Dy-Snake environment, which is a discrete control environment, we present the results of the chasing food task in Figure 6. The results show that the proposed algorithm works best. DQN is better than HER. HER fails for this task. That HER fails for this task shows just using achieved goals is not enough for the tasks with dynamic goals and can lead wrong direction. The results also show that around $800$ episodes, the performance of DQN and DHER is close and DHER is slightly better than DQN. This is because the chasing food task is a simple task. The action space is very small and just $4$ types of actions. After enough exploration, DQN also has competitive performance. However, at the beginning, DHER quickly achieves very good performance. It shows that assembling experience from two failures improves the performance very efficiently in this task. DQN (dense) and DQN (dense-2) help learn the policy at the early stage. However, in the long run, it does not lead to any particular benefits.

### 4.5 SIM TO REAL ADAPTATION

We used policies for Dy-Circling task and a new Pouring task trained in our simulator [3] to deploy them on a physical robot. As shown in Figure 8, the policies were trained by using DHER and adapted to the real robot without any finetuning. However, the policy requires accurate localizations of the gripper and the goal. For Dy-Circling task, the robot's gripper was blocked. There were a toy turntable, whose speed is unknown, and a blue block on the turntable. We set the position 1cm above the block as the target position. The position of the block was predicted based on traditional contour shape analysis using camera images. For Pouring task, the robot gripped a can. A man held a cup and moved it. The cup was set as the target and with a green marker. We used the marker to estimate its position.

Our policies were transferred successfully for both tasks. With the accurate positions, we have 100% success rate for 5 trials. It was observed that the robot had learned to not only follow the current target but also step forward to the future target position. Demo videos about the experiments are available at `https://sites.google.com/view/dher`.

---

[3]We developed a new simulator according to our hardware. More details on our hardware setup are available in the appendix.

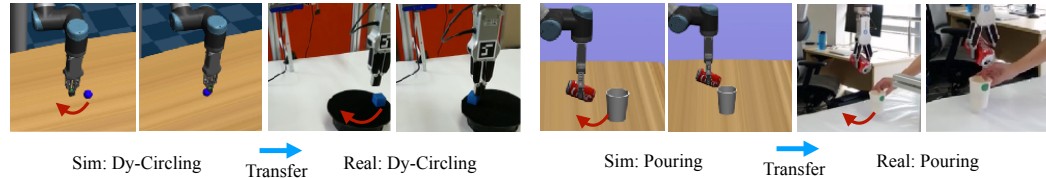

Sim: Dy-Circling    Transfer    Real: Dy-Circling        Sim: Pouring    Transfer    Real: Pouring

Figure 8: Adapting the policies trained based on DHER from our simulation to a real robotic arm.

### 4.6 SPECIAL CASE: HOW ABOUT IF HER KNOWS THE LAW OF THE MOTION OF A GOAL

We use Dy-Snake to demonstrate the experimental results for a special case that the law of the motion of the target (food) is known to agents. Because the motion of the food is very simple and controllable in Dy-Snake environment. We develop a direct extension of HER, called HER+, that modifies desired goals at every timestep based on the law of the motion of the food to create successful experience. More details of HER+ are described in the appendix.

We show the results in Figure 7. DHER and HER+ are both better than DQN at the beginning. DHER is slightly better than HER+, which shows the efficiency of DHER is comparable in this simple task.

## 5 CONCLUSION

We introduced a novel technique that assembles successful experience from a couple of failures. With this technique, our proposed algorithm called DHER ( Dynamic Hindsight Experience Replay) is able to address the tasks with sparse rewards and dynamic goals. Our technique can be combined with an arbitrary off-policy RL algorithm and we experimentally demonstrated that with DQN and DDPG. As far as we know, it is the first time that an agent is allowed to learn from assembled experience from two failures.

## ACKNOWLEDGMENTS

We would like to thank Weitao Xi, Tianzhou Wang, Tingguang Li for performing some additional transfer experiments. We would also like to thank Han Liu and the whole RL team for fruitful discussions as well as the anonymous reviewers for their comments.

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

APPENDIX

TASKS WITH DYNAMIC GOALS

We have created four tasks as described below and illustrated by Figure 2. The first three tasks are based on the robotic environments. The last is based on the GREEDY SNAKE environment.

- Dy-Reaching: The task is to control the robotic arm so that its gripper can reach the target position. The target position moves from one point to another along a straight line with a constant velocity.
- Dy-Circling: The task is to control the robotic arm so that its gripper can reach the target position. The target position moves along a circle with a constant velocity.
- Dy-Pushing: in this task, a box is placed on a table in front of the robotic arm. The robot is required to move the box to the target location on the table. The target location moves from one position to another with a fixed velocity. Note that the robot's grippers are locked to prevent it from grasping. The learned behavior is actually a mixture of pushing and rolling.
- Dy-Snake: in this task, a snake and food are placed on a rectangle map and the task is to control the snake to eat the food. The food is moving from one position to another position with a fixed velocity.

HARDWARE

We use the Universal Robots UR10 with a gripper. We use a RealSense Camera SR300 to track the position of objects. The gripper is blocked. We use a marker on the gripper for camera calibration. During adapting the robot, for the position of the target, we use the camera to estimate it.

SIMULATION

We simulate the physical system using the MuJoCo physics engine (Todorov et al., 2012) and also use MuJoCo to render the images. In our tasks, we use positions provided by MuJoCo for training policies in the simulation.

In Figure 9, we demonstrate our simulation and the physics environment.

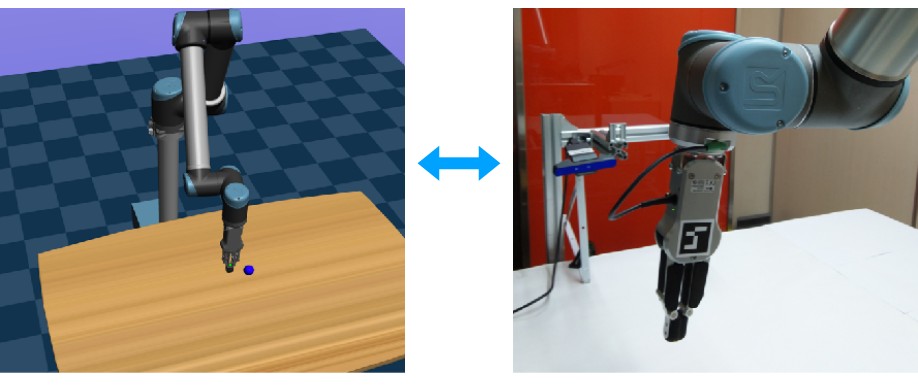

**Simulation**                    **Physics environment**

Figure 9: Our simulation and the physics environment.

A SPECIAL AND SIMPLER CASE - HER+

The paper considers a general situation that the law of the motion of goals is invisible to an agent. However, we relax this assumption and assume that in some situations the law of the motion of goals

is easy to obtain, i.e., the parameters of $g(t; \gamma)$ are known. For example, it is able to calculate the velocity of an object by observing the trajectories of the object.

In this situation, we do not need to search failed experience. With the knowledge of the velocity of an object, it is straightforward that we can calculate any trajectory of desired goals. Thus for every failed experience, at any time step $t$, based on the achieved goal $g_t^{ac}$, we calculate new desired goals $g_t^{de} = g(t)$ corresponding to the achieved goals in order to construct successful experience.

The details of HER+ are described in Algorithm 2. HER+ can be seen as a direct extension of HER. HER and HER+ can modify any failed experience to successful experience because they both assume the law of the motion of goals is known.

---

**Algorithm 2** Hindsight Experience Replay Plus

---
**Require:** an off-policy RL algorithm $\mathbb{A}$, replay buffer $R$, a reward function $r$
 1: Initialize $\mathbb{A}$ and replay buffer $R$
 2: **for** episode $= 1, 2, \cdots, M$ **do**
 3:     Sample an initial goal $g_0$ and an initial state $s_0$
 4:     **for** $t = 0, \cdots, T-1$ **do**
 5:         Sample an action $a_t$ using the behavioral policy from $\mathbb{A}$:
 6:         $a_t \leftarrow \pi(s_t|g_t)$
 7:         Execute the action $a_t$ and observe a new state $s_{t+1}$ and a new goal $g_{t+1}$
 8:     **end for**
 9:     **for** $t = 0, \cdots, T-1$ **do**
10:         $r_t := r(s_t, a_t, g_{t+1})$
11:         Store the transitions $(s_t|g_t, a_t, r_t, s_{t+1}|g_{t+1})$ in $R$   (Standard experience replay)
12:         Sample a set of the achieved goals of $E_i$ as additional goals for reply $G'$
13:         **for** $g_p' \in G'$ **do**
14:             Calculate a goal trajectory $\{g_0', \cdots, g_p'\}$ where $g_t' = g(t; \gamma)$   (HER+)
15:             **for** $t = \{0, \cdots, p-1\}$ **do**
16:                 $r_t' := r(s_{i,t}, a_{i,t}, g_{t+1}')$
17:                 Store the transition $(s_{i,t}|g_t', a_{i,t}, r_t', s_{i,t+1}|g_{t+1}')$ in $R$
18:             **end for**
19:         **end for**
20:     **end for**
21: **end for**

---

