# OpenReview forum: "DHER: Hindsight Experience Replay for Dynamic Goals"
_ICLR.cc/2019/Conference_

### Official Review · AnonReviewer2 · 2018-10-22
**Simple and nice idea, but very unclear description and some serious flaws**

**Rating:** 6
**Confidence:** 3

**Review:**

In this paper, the authors extend the HER framework to deal with dynamical goals, i.e. goals that change over time.
In order to do so, they first need to learn a model of the dynamics of the goal, and then to select in the replay buffer experience reaching the expected value of the goal at the expected time. Empirical results are based on three (or four, see the appendix) experiments with a Mujoco UR10 simulated environment, and one experiment is successfully transfered to a real robot.

Overall, the addressed problem is relevant (the question being how can you efficiently replay experience when the goal is dynamical?), the idea is original and the approach looks sound, but seems to suffer from a fundamental flaw (see below).

Despite some merits, the paper mainly suffers from the fact that the implementation of the approach described above is not explained clearly at all.
Among other things, after reading the paper twice, it is still unclear to me:
- how the agent learns of the goal motion (what substrate for such learning, what architecture, how many repetitions of the goal trajectory, how accurate is the learned model...)
- how the output of this model is taken as input to infer the desired values of the goal in the future: shall the agent address the goal at the next time step or later in time, how does it search in practice in its replay buffer, etc.

These unclarities are partly due to unsufficient structuring of the "methodology" section of the paper, but also to unsufficient mastery of scientific english. At many points it is not easy to get what the authors mean, and the paper would definitely benefit from the help of an experienced scientific writer.

Note that Figure 1 helps getting the overall idea, but another Figure showing an architecture diagram with the main model variables would help further.

In Figures 3a and 5, we can see that performance decreases. The explanation of the authors just before 4.3.1 seem to imply that there is a fundamental flaw in the algorithm, as this may happen with any other experiment. This is an important weakness of the approach.

To me, Section 4.5 about transfer to a real robot does not bring much, as the authors did nothing specific to favor this transfer. They just tried and it happens that it works, but I would like to see a discussion why it works, or that the authors show me with an ablation study that if they change something in their approach, it does not work any more.

In Section 4.6, the fact that DHER can outperform HER+ is weird: how can a learn model do better that a model given by hand, unless that given model is wrong? This needs further investigation and discussion.

In more details, a few further remarks:

In related work, twice: you should not replace an accurate enumeration of papers with "and so on".

p3: In contrary, => By contrast,

which is the same to => same as

compare the above with the static goals => please rephrase

In Algorithm 1, line 26: this is not the algorithm A that you optimize, this is its critic network.

line 15: you search for a trajectory that matches the desired goal. Do you take the first that matches? Do you take all that match, and select the "best" one? If yes, what is the criterion for being the best?

p5: we find such two failed => two such failed

that borrows from the Ej => please rephrase

we assign certain rules to the goals so that they accordingly move => very unclear. What rules? Specified how? Please give a formal description.

For defining the reward, you use s_{t+1} and g_{t+1}, why not s_t and g_t?

p6: the same cell as the food at a certain time step. Which time step? How do you choose?

The caption of Fig. 6 needs to be improved to be contratsed with Fig. 7.

p8: the performance of DQN and DHER is closed => close?

DHER quickly acheive(s)

Because the law...environment. => This is not a sentence.

Mentioning in the appendix a further experiment (dy-sliding) which is not described in the paper is of little use.

---

> ### Author Response · Authors · 2018-11-09
> **[2/2] Additional responses to the reviewer’s points.**
>
> Q4: an architecture diagram
> A4: We updated Figure 1.
>
> Q5: Figures 3a and 5 … performance decreases ...
> A5: One reason may be that it is a temporal drop and will recover later. Another reason may be that the policy trained with assembled experiences becomes overfitting to simple cases as such kind of experiences are assembled a lot. The overfitting to simple cases decreases overall performance. The similar pattern also appeared in other papers. See Pusing task in Fig 2 in HER (Andrychowicz et al., 2017).
>
> Q6: To me, Section 4.5 about transfer to a real robot does not bring much …
> A6: The experiments of transferring to a real robot mainly demonstrate dynamic goals are real-world problems and can be solved by our method. At the same time, it shows when DHER uses positions, it is robust to the real-world environment.
>
> Q7: In Section 4.6, the fact that DHER can outperform HER+ is weird …
> A7: It is indeed a little surprising. It shows DHER is very efficient in some simple environments. In a simple environment, such as Dy-Snake, DHER has better generation than HER+. The reason may be that HER+ uses only one way to modify a trajectory. However, DHER has different ways to create success trajectories because we can find different matching positions given a trajectory from the past experiences. The Dy-Snake environment is so simple that DHER is able to create a lot of success experience in a short time.
>
> Q8: In more details, a few further remarks ...
> A8: We polished the paper.
>
> Q9: in the appendix a further experiment (dy-sliding) … of little use…
> A9: We removed it. We added this before because our open source will contain this environment and our model also works on it successfully.
>
> Q10: In Algorithm 1, line 26: this is not the algorithm A that you optimize, this is its critic network.
> A10: Line 26 indicates a standard update for the RL algorithm A. It is similar to HER. Please see the last several lines of Algorithm 1 in HER (Andrychowicz et al., 2017).
> The key process of DHER is from lines 13 to 23. We had added a marker at the end of Line 20.
>
> Q11: line 15: you search for a trajectory that matches the desired goal ...
> A11: We use a hash table to store trajectories. We search trajectories in the hash table and return the first that matches.
>
> Q12: we assign certain rules to the goals so that they accordingly move => very unclear...
> A12: The details are given in the next paragraph. See the second paragraph in Section 4.1. For different environments, the rules are slightly different.
>
> Q13: For defining the reward, you use s_{t+1} and g_{t+1}, why not s_t and g_t?
> A13: They are the same meaning and just corresponding to different timesteps. At time step t, after taking an action, the state turns to s_{t+1} and the goal turns to g_{t+1}. Thus the reward is defined based on s_{t+1} and g_{t+1}.
> Similarly, if the time step is t - 1 (t > 1), the reward is defined based on s_{t} and g_{t}.
>
> Q14: p6: the same cell as the food at a certain time step. Which time step? How do you choose?
> A14: It means if the snake moves to the same cell as the food at any timestep, the game is over. We only set the maximum timestep for each episode.

---

> > ### Comment · AnonReviewer2 · 2018-11-23
> > **I still don't get it**
> >
> > I'm really sorry, but after reading Section 3 several times, I'm afraid I still don't understand exactly what the authors are doing.
> >
> > Let us take the example of a moving object that the agent must reach. In this case, I'm assuming "acheived goals" correspond to positions of the object, known a posteriori. But what do "desired goals" stand for? Is the agent trying to predict the trajectory of the object?
> >
> > It seems that the other reviewers have no problem with that, so if anybody could explain the setup to me, I would be delighted...

---

> > > ### Author Response · Authors · 2018-11-23
> > > **More details of the desired and achieved goals**
> > >
> > > Thanks for your reply. We follow the goal setting of UVFA (Schaul et al., 2015a) and HER (Andrychowicz et al., 2017). In the example of a moving object that the agent must reach, the desired goals correspond to the positions of the moving object. The achieved goals correspond to the positions of the gripper, which is connected to a robotic arm.
> > > The moving object (desired goals) is a dynamic goal. The gripper (achieved goals) is controlled by the RL agent and tries to reach the moving object.
> > >
> > > We updated Figure 1 and Section 3.2  to add more details of the desired and achieved goals. In the paper, in Figure 1, we use a green curve to indicate the trajectory of achieved goals and a red curve to indicate the trajectory of desired goals.
> > > In Figure 2, for the first three tasks, the positions of the (blocked) black gripper indicate achieved goals. The red object indicates desired goals. For the fourth task, the green circle (a snake) indicates achieved goals. The red object (food) indicates desired goals.
> > >
> > > In our real robotic system, there are a gripper connected to a robotic arm and a blue block. When running a task, the positions of the gripper indicate achieved goals and the positions of the moving blue block indicate desired goals.

---

> > > > ### Comment · AnonReviewer2 · 2018-11-23
> > > > **Goal trajectory prediction ?**
> > > >
> > > > Thank you for the clarification above, it helps a lot. Now, to catch a moving object, you have to predict one of its future positions to meet it on time. You are not learning a model of the dynamics. So how do you predict this future position? Is it based on the idea that object trajectories are repeatable, i.e. the same object will perform the same trajectory many times?
> > > >
> > > > Please forgive this naive way of putting questions, but it may help many readers beyond me.

---

> > > > > ### Author Response · Authors · 2018-11-23
> > > > > **No goal trajectory prediction but use RL to predict actions**
> > > > >
> > > > > Thanks for your reply. We do not directly predict or model the future of desired goals. Given a lot of the past trajectories of desired goals, corresponding actions, and rewards (which can be seen as scores for taken actions), the RL agent will know which action is good and can lead to the desired goals in the future.
> > > > > The RL algorithm constructs a policy (normally a neural network) to determine how to choose an action based on the current state. Therefore, we can conclude that the prediction of desired goals is automatically embedded in the learned RL policy. It is achieved from the process of maximizing the rewards by the RL agent.

---

> > > > > ### Public Comment · (anonymous) · 2018-11-26
> > > > > **On the repeatable object trajectory**
> > > > >
> > > > > As long as there are some patterns in the desired goal's motion, the proposed method should work. It means that it doesn't have to follow the same trajectory. Of course, when the pattern is simple, it becomes easy to learn.

---

> ### Author Response · Authors · 2018-11-09
> **[1/2] The algorithm does not need to learn the dynamics. It creates success experiences by combining trajectories in the replay buffer.**
>
> We thank the reviewer for the comments and have revised the paper accordingly. We believe the reviewer has some misunderstandings about our work. We make the following clarifications.
> 1) For the dynamic of goals, our algorithm does not need to learn the dynamics. The algorithm creates new experiences through combining two failed experiences in which their goal trajectories have overlaps at any timestep. Please see paragraph 1 in Section 3.1 (Dynamic goals) for more descriptions.
> 2) Our algorithm is about experience replay. The input is the past experiences. The output is new assembled success experiences if exist. We updated Figure 1 to show how DHER works with a RL algorithm.
> 3) Regarding the RL environments and the proposed algorithm and transfer solution, we would like to open all of them. All results can be reproduced. We believe the dynamic goal problem manipulation control is also interesting for other researchers.
>
>
> Q1: In order to do so, they first need to learn a model of the dynamics of the goal, and then to select in the replay buffer experience reaching the expected value of the goal at the expected time.
> A1: Please see S1.
>
> Q2: how the agent learns of the goal motion ...
> A2: Generally speaking, reinforcement learning learns a policy through trial and error. The reinforcement learning agent interacts with an environment and obtains rewards to indicate whether its action is good or not.
> In our setting, the goal’s motion is a part of environment. This setting is quite normal in real world. See our introduction and HER (Andrychowicz et al., 2017). When a RL algorithm takes an action, it will automatically and latently take the knowledge of the goal’s motion into consideration.
> However, under this setting, after interacting with the environment for a long time, we still face the problem that we do not have success signals to guide a policy learning. The main difficulty then lies in how to efficiently use the past experiences in the replay buffer to construct the success signals, other than learn the motion of the goal. Our paper then provide a solution to solve the difficulty.
> There are a lot of goal trajectories and they are different to each other. Taking Dy-Reach as an example, as shown in Figure 3(a), we followed openai gym’s training settings. There are 50 epoches in total and each epoch has 100 episodes, i.e,. 100 trajectories. The performance (success rate) of the learned model DDPG+DHER can achieve 0.8. If the velocity of the goal is slower, the performance can achieve 1.0.
>
> Q3: how the output of this model is taken as input to infer the desired values of the goal in the future: ...
> A3: Our model is a kind of experience reply method. The input of our model is the past trajectories. Most of them are failed. The output of our model is assembled experiences. The assembled experiences are success experiences. We followed the goal setting of UVFA (Schaul et al., 2015a) and HER (Andrychowicz et al., 2017). The goals are represented by positions. The model searches the relevant experience according to the positions of goals. If the positions of two goals are overlapped (< tolerance 0.01) at anytime, then they are matched.

---

### Official Review · AnonReviewer3 · 2018-11-02

**Rating:** 7
**Confidence:** 4

**Review:**

This paper proposes a way of extending Hindsight Experience Replay (HER) to dynamic or moving goals. The proposed method (DHER) constructs new successful trajectories from pairs of failed trajectories where the goal accomplished at some point in the first trajectory happens to match the desired goal in the second trajectory. The method is demonstrated to work well in several simulated environments and some qualitative sim2real transfer results to a real robot are also provided.

The paper is well written and is mostly easy to follow. I liked the idea of combining parts of two trajectories and to the best of my knowledge it is new. It is a simple idea that seems to work well in practice. While DHER has some limitations I think the key ideas will lead to interesting future work.

The main shortcoming of the paper is that it does not consider other relevant baselines. For example, since the position of the goal is known, why not use a shaped reward as opposed to a sparse reward? The HER paper showed that using sparse rewards with HER can work better than shaped rewards. These findings may or may not transfer to the dynamic goal case so including a shaped reward baseline would make the paper stronger.

Some questions and suggestions on how to improve the paper:
- It would be good to be more upfront about the limitations of the method. For example, the results on a real robot probably require accurate localization of the gripper and cup. Making this work for precise manipulation will probably require end-to-end training from vision where it’s not obvious DHER would apply.
- It would be interesting to see quantitative results for the simulated experiments in section 4.5.
- The performance of DHER on Dy-Reaching seems to degrade in later stages of training (Figures 3a and 5). Do you know what is causing it? DQN or DHER?

Overall, I think this a good paper.

---

> ### Author Response · Authors · 2018-11-09
> **Experiments of the shaped reward baselines.**
>
> Thank you for your insightful comments and feedback!
>
> Q1: baselines … shaped rewards…
> A1: We added shaped reward baselines. We use a natural distance related (dense) reward function to train the agent. Figures 3 and 6 in the paper show that the dense rewards do not work well for dynamic goals, though they help at the beginning of the learning.
>
> Q2: - It would be good to be more upfront about the limitations of the method …
> A2: We agree. In the revised paper, we provided more details about the limitations, including the goal assumption, the transfer requirements and so on. See Section 1 and 4.5 for more details.
>
> Q3: It would be interesting to see quantitative results for the simulated experiments in section 4.5.
> A3: Thanks for your valuable suggestion. In Section 4.5, with the accurate positions, we have 100% success rate for 5 trials.
>
> Q4: The performance of DHER on Dy-Reaching seems to degrade in later stages of training (Figures 3a and 5). Do you know what is causing it? DQN or DHER?
> A4: One reason may be that it is a temporal drop and will recover later. Another reason may be that the policy trained with assembled experiences becomes overfitting to simple cases as such kind of experiences are assembled a lot. The overfitting to simple cases decreases overall performance. The similar pattern also appeared in other papers. See Pusing task in Fig 2 in HER (Andrychowicz et al., 2017).

---

### Official Review · AnonReviewer1 · 2018-11-02
**Interesting idea but lacking some context and experiments seem to not have any  baselines targeted at the problem**

**Rating:** 6
**Confidence:** 4

**Review:**

The authors propose an extension of hindsight replay to settings where the goal is moving. This consists in taking a failed episode and constructing a valid moving goal by searching prior experiences for a compatible goal trajectory. Results are shown on simulated robotic grasping tasks and a toy task introduced by the authors. Authors show improved results compared to other baselines. The authors also show a demonstration of transfering their policies to the real world.

The algorithm appears very specific and not applicable to all cases with dynamic goals. It would be good if the authors discussed when it can and cannot be applied. My understanding is it would be hard to apply this when the environment changes across episodes as there needs to be matching trajectories. It would also be hard to apply  this for the same reason if there are dynamics chaging the environment (besides the goal). If the goal was following more complex dynamics like teleporting from one place it seems it would again be rather hard to adapt this. I am also wondering if for most practical cases one could construct a heuristic for making the goal trajectory a valid one (not necessarily relying on knowing exact dynamics) thus avoiding the matching step.

The literature review and the baselines do not appear to consider any other methods designed for dynamic goals. The paper seems to approach the dynamic goal problem as if it was a fresh problem. It would be good to have a better overview of this field and baselines that address this problem as it has certainly been studied in robotics, computer vision, and reinforcement learning. I find this paper hard to assess without a more appropriate context for this problem besides a recently proposed technique for sparse rewards that the authors might want to adapt to it.  I find it difficult to believe that nobody has studied solutions to this problem and solutions specific to that don’t exist.

The writing is a bit repetitive at times and I do believe the algorithm can be more tersely summarized earlier in the paper. It’s difficult to get the full idea from the Algorithm block.

Overall, I think the paper is borderline. There is several interesting ideas and a new dataset introduced, but I would like to be more convinced that the problems tackled are indeed as hard as the authors claim and to have a better literature review.

---

> ### Author Response · Authors · 2018-11-09
> **There is little work addressing dynamic goals in the sparse reward setting. Update the literature review and add dense reward baselines.**
>
> We thank the reviewer for the comments, and we would like to clarify a few important misconceptions that the reviewer has regarding our work.
> 1) We position the paper in the context of RL with sparse rewards. We follow the goal setting of UVFA (Schaul et al., 2015a) and HER (Andrychowicz et al., 2017). The dynamic goal problem is extended from this setting, not all other cases. Please see paragraph 3 in Section 1 (Introduction) and paragraph 1 in Section 3.1 (Dynamic goals) for more descriptions.
> 2) We propose a new experience replay method. The proposed algorithm can be combined with any off-policy RL algorithms, similar to HER, as shown in Figure 1.
> 3) The motivation of developing algorithms which can learn from unshaped reward signals is that it does not need domain-specific knowledge and is applicable in situations where we do not know what admissible behaviour may look like. The similar motivation is also mentioned in HER (Andrychowicz et al., 2017). We also added new experimental results about dense rewards. The results show DHER works better. See Figures 3 and 6.
>
> Q1: The algorithm appears very specific and not applicable to all cases with dynamic goals. …
> A1: Please see 1) and 3) above.
>
> Q2: I am also wondering if for most practical cases one could construct a heuristic for making the goal trajectory a valid one (not necessarily relying on knowing exact dynamics) thus avoiding the matching step.
> A2: It is a good idea to take domain heuristics into consideration. However, in our paper, we aim to construct a model-free method for dynamic goals to avoid the complexity of constructing goal trajectories. We agree that your idea worths a try in the future.
>
> Q3: The literature review and the baselines do not appear to consider any other methods designed for dynamic goals. …
> A3: We do not want to claim that the dynamic goal problem is a fresh problem. However, there is little work addressing dynamic goals in the sparse reward setting. As far as we know, there is no open-source RL environments for such problems. (OpenAI Gym Robotics uses fixed goals.)
>
> Q4:  I find it difficult to believe that nobody has studied solutions to this problem and solutions specific to that don’t exist.
> A4: Our paper focuses on addressing dynamic goals with sparse rewards. This setting has not been addressed probably because it is difficult to learn. For example, the recently developed DDPG and HER failed in our tasks. Moreover, there is no open-source environments for the dynamic goals and sparse rewards, to the best of our knowledge.
>
> Q5: There is several interesting ideas and a new dataset introduced, but I would like to be more convinced that the problems tackled are indeed as hard as the authors claim and to have a better literature review.
> A5: Except for sparse rewards, we also added new experimental results about dense rewards for the dynamic goal setting. We have similar results. Similar to DDPG and DDPG+HER, DDPG(dense) does not work well in our tasks. For the simple Dy-Snake environment, DQN(dense) is better than DQN but not better than DQN+DHER. See Figures 3 and 6.

---

> > ### Comment · AnonReviewer1 · 2018-11-09
> > **The approach appears more problem specific than claimed.**
> >
> > Thanks for your response and clarifications. I would like to comment on this point:
> >
> > "1) We position the paper in the context of RL with sparse rewards. We follow the goal setting of UVFA (Schaul et al., 2015a) and HER (Andrychowicz et al., 2017). The dynamic goal problem is extended from this setting, not all other cases. Please see paragraph 3 in Section 1 (Introduction) and paragraph 1 in Section 3.1 (Dynamic goals) for more descriptions.
> > 2) We propose a new experience replay method. The proposed algorithm can be combined with any off-policy RL algorithms, similar to HER, as shown in Figure 1.
> > 3) The motivation of developing algorithms which can learn from unshaped reward signals is that it does not need domain-specific knowledge and is applicable in situations where we do not know what admissible behaviour may look like. The similar motivation is also mentioned in HER (Andrychowicz et al., 2017). We also added new experimental results about dense rewards. The results show DHER works better. See Figures 3 and 6.
> >
> > Q1: The algorithm appears very specific and not applicable to all cases with dynamic goals. …
> > A1: Please see 1) and 3) above."
> >
> > I believe this kind of motivation as a principled approach to RL  with sparse rewards and no domain knowledge is an overclaim. The HER algorithm is a heuristic one and to the best of my understanding requires a domain specific knowledge of how to set fake goals, which is natural in many settings such as grid worlds for example. The moving goal case described here requires even more domain specific knowledge and I am not convinced is truly “model-free” in most cases.  To the best of my understanding the matching phase of your method requires a domain specific understanding of goal similarity. Is it possible to provide a dynamic goal example that is not just a simple and short trajectory in space and makes sense to be applied with DHER? Could the authors for example explain how the algorithm would be applicable in a case of an Atari style game where a goal would teleport or have long trajectories (non-trivial to match without a complex matching heuristic). It seems in this case (a) one would have to obtain precise coordinate positions of the goal (this would mean one can’t just solve the problem based on pure pixels and must rely on domain knowledge) and (b) the matching algorithm itself would need to be heavily crafted with domain specific knowledge. I think the method might be more specific than the authors claim and should be presented as such.

---

> > > ### Author Response · Authors · 2018-11-13
> > > **Limitation and that the algorithm is very natural for many manipulation tasks**
> > >
> > > Thanks for discussing the limitation of DHER. Similar to HER, we need to have the definition of goals and know the similarity metric between goals in order to construct “success” from failed experiences. We had provided how to use and define goals in Section 3.1 --- and we made addition revisions to make it more clear. See Sections 1 and 3.1 for the discussions.
> > >
> > > Because we have the same multi-goal assumption as HER, we did not claim our method can be used for every case. However, it still can be applied to many domains if we know how to define the goals and if their trajectories intersect.
> > >
> > > For a game, if its goals can be used as part of the observation and do not affect the environment dynamics, our algorithm will work. Regarding the Atari games, we did find that there is no game satisfying the multi-goal assumption. However, our approach can be potentially used for other games where we know the similarity of goals, for example, hunting for food in a Minecraft-like grid world mini-game. The Dy-Snake game in our work serves as a reference for which types of games our approach can benefit.
> > >
> > > The algorithm is very natural for many manipulation tasks because we can access (sometimes noisy) object positions in manipulation. The starting point of this work is actually for manipulation controls.

---

### Public Comment · ~Hassam_Sheikh1 · 2018-11-21
**More suitable for a robotics conference**

I feel this paper is very well suited for a robotics conference rather than ICLR, I see valid concerns from reviewers but these are  general assumptions that needs to be taken by robotics people to make things work.  The only thing, I am interested in the overall time-complexity of DHER process, which seems to be growing as the number of episodes increases

---

> ### Author Response · Authors · 2018-11-22
> **The topic of the paper and the time complexity of DHER**
>
> Thanks for your interest. The paper belongs to relevant topics in ICLR, for example, reinforcement learning or applications in robotics, or any other field. Please see https://iclr.cc/Conferences/2019/CallForPapers .
> Take HER (Andrychowicz et al., 2017) as an example, it was published in NIPS 2017.
>
> For DHER, the time complexity of search process is O(1). In our implementation, we use two hash tables to store the trajectories of achieved goals and desired goals, respectively.

---

### Author Response · Authors · 2019-01-03
**Code Release**

The code includes environments and algorithms: https://github.com/mengf1/DHER .

---

### Public Comment · ~Xiaojian_Ma1 · 2019-02-15
**Some questions**

I found this paper and the supplementary video are quite interesting and insightful. However, I got a few questions on some details, and hope you can help me with them.

Q1: In the last few seconds of  video#1(https://drive.google.com/file/u/0/d/10SU6vYd0m0MAtSpRbs71sTQ4G6gaoBnY/view), when the gripper strikes out the gray ball, the red ball(goal) still locates on the very left of the table. According to my understanding, the goal at that time will be striking the gray ball to reach the very left position. However, the arm chooses to strike the ball to the middle instead. It's quite non-intuitive to observe such results since I can't find any mechanism about goal prediction in DHER and its implementation. Furthermore, if it can be seen as some mysterious generalization of the trained policy, why the policies in other Dy-* tasks are mostly acting in a goal-following manner? Can you help me to figure it out?

Q2: In DHER, it matches the desired goal traj and achieved goal traj to rewrite the goal information of the off-policy experience. An emerging problem will be when the feasible space is quite large, it may require a pretty large amount of trajs to make a successful match, which will hurt the sample efficiency of such off-policy methods (DPG). I wonder if it is possible to simply construct a desired goal traj manually instead of brute-force matching the simulated data? The temporal alignment may be an issue, but as using simulation is allowed (also used in the matching procedure of DHER), I think it won't be difficult.

---

### Meta-Review · Area_Chair1 · 2018-12-16
**A potentially influential approach despite its limitations, well delivered and improved following feedback.**

**Confidence:** 4
**Recommendation:** Accept (Poster)

**Metareview:**

This work proposes a method for extending hindsight experience replay to the setting where the goal is not fixed, but dynamic or moving. It proceeds by amending failed episodes by searching replay memory for a compatible trajectories from which to construct a trajectory that can be productively learned from.

Reviewers were generally positive on the novelty and importance of the contribution. While noting its limitations, it was still felt that the key ideas could be useful and influential. The tasks considered are modifications of OpenAI robotics environments, adapted to the dynamic goal setting, as well as a 2D planar "snake" game. There were concerns about the strength of the baselines employed but reviewers seemed happy with the state of these post-revision. There were also concerns regarding clarity of presentation, particularly from AnonReviewer2, but significant progress was made on this front following discussions and revision.

Despite remaining concerns over clarity I am convinced that this is an interesting problem setting worth studying and that the proposed method makes significant progress. The method has limitations with respect to the sorts of environments where we can reasonably expect it to work (where other aspects of the environment are relatively stable both within and across episodes), but there is lots of work in the literature, particularly where robotics is concerned, that focuses on exactly these kinds of environments. This submission is therefore highly relevant to current practice and by reviewers' accounts, generally well-executed in its post-revision form. I therefore recommend acceptance.